# The American Cherimoya Genome Reveals Insights into the Intra-Specific Divergence, the Evolution of *Magnoliales*, and a Putative Gene Cluster for Acetogenin Biosynthesis

**DOI:** 10.3390/plants13050636

**Published:** 2024-02-26

**Authors:** Tang Li, Jinfang Zheng, Orestis Nousias, Yuchen Yan, Lyndel W. Meinhardt, Ricardo Goenaga, Dapeng Zhang, Yanbin Yin

**Affiliations:** 1Nebraska Food for Health Center, Department of Food Science and Technology, University of Nebraska, Lincoln, NE 68588, USA; tang.li@huskers.unl.edu (T.L.); zhengjinfang1220@gmail.com (J.Z.); orestis.nousias@yale.edu (O.N.); yyan15@huskers.unl.edu (Y.Y.); 2Sustainable Perennial Crops Laboratory, United States Department of Agriculture, Agriculture Research Service, Beltsville, MD 20705, USA; lyndel.meinhardt@usda.gov; 3Tropical Agriculture Research Station, United States Department of Agriculture, Agriculture Research Service, Mayaguez 00680, Puerto Rico; ricardo.goenaga@usda.gov

**Keywords:** *Annona*, *Annonaceae*, *Magnoliales*, whole genome duplication, Neotropical region, tropical Americas, domestication, diversity

## Abstract

*Annona cherimola* (cherimoya) is a species renowned for its delectable fruit and medicinal properties. In this study, we developed a chromosome-level genome assembly for the cherimoya ‘Booth’ cultivar from the United States. The genome assembly has a size of 794 Mb with a N50 = 97.59 Mb. The seven longest scaffolds account for 87.6% of the total genome length, which corresponds to the seven pseudo-chromosomes. A total of 45,272 protein-coding genes (≥30 aa) were predicted with 92.9% gene content completeness. No recent whole genome duplications were identified by an intra-genome collinearity analysis. Phylogenetic analysis supports that eudicots and magnoliids are more closely related to each other than to monocots. Moreover, the *Magnoliales* was found to be more closely related to the Laurales than the Piperales. Genome comparison revealed that the ‘Booth’ cultivar has 200 Mb less repeats than the Spanish cultivar ‘Fino de Jete’, despite their highly similar (>99%) genome sequence identity and collinearity. These two cultivars were diverged during the early Pleistocene (1.93 Mya), which suggests a different origin and domestication of the cherimoya. Terpene/terpenoid metabolism functions were found to be enriched in *Magnoliales*, while TNL (**T**oll/Interleukin-1-**N**BS-**L**RR) disease resistance gene has been lost in *Magnoliales* during evolution. We have also identified a gene cluster that is potentially responsible for the biosynthesis of acetogenins, a class of natural products found exclusively in *Annonaceae*. The cherimoya genome provides an invaluable resource for supporting characterization, conservation, and utilization of *Annona* genetic resources.

## 1. Introduction

Cherimoya (*Annona cherimola*) is a popular fruit widely cultivated in tropical and subtropical regions [1,2]. *A. cherimola* (2n = 2x = 14) belongs to the *Annona* genus in the *Annonaceae* family, which is the largest family in the *Magnoliales* order [3]. This family contains around 108 genera and 2400 known species. In addition to *A. cherimola*, there are several other *Annona* species with edible fruits, including *Annona muricata*, *Annona squamosa*, *Annona reticulata*, *Annona purpurea* and the interspecific hybrid Atemoya (*A. cherimola* × *A. squamosa*) [1]. Cherimoya fruits are commonly eaten fresh. The fruit is very sweet, with a custard-like texture. The aromatic flavor is a mix of pineapple, mango, and strawberry [4]. Cherimoya can also be used to make a range of food products, including ice creams, milkshakes, jellies, yogurt, juice, and wine [5]. The fruit is a good source of vitamin C and vitamin B, fiber, flavonoids, and potassium [6,7].

Besides its utilization as a fruit, cherimoya has long been used as traditional medicine in different American civilizations. It provides various health benefits such as promoting digestion, preventing high blood pressure and supporting immunity [8,9]. This fruit has attracted special attention in recent years due to its high concentration of acetogenins—a class of lipophilic polyketide natural products uniquely found in *Annonaceae* species [10,11,12]. Research has accredited beneficial effects to acetogenins, including the induction of cytotoxic, anti-inflammatory and anti-tumorous activities [13,14,15]. The chemical characteristics, bioactivity and achievements regarding the therapeutic usage of acetogenins from cherimoya were recently reviewed [16].

Cherimoya is indigenous to the tropical Americas [17,18]. However, the precise center of origin of cherimoya is still controversial. The early chroniclers and scientists proposed that the native home of cherimoya is in the inter-Andean valleys, including northern Peru and southern Ecuador [5,17]. This hypothesis was supported by linguistic data, botanical remains and ceramics of Chimu-Inca and Moche cultures in northern Peru, proving the presence of cherimoya in Peru as early as 2500 B.C. [19]. The existence of natural populations of cherimoya in the mountain valleys in southern Ecuador and northern Peru provides additional evidence to support this hypothesis [17,20]. However, more recent studies based on microsatellite markers revealed a higher allelic diversity in Mesoamerican cherimoya germplasm compared to the diversity found in South America [18,21]. These results suggested the Mesoamerican origin of cherimoya and a pre-Columbian movement of plant material, probably seeds, to South America, which resulted in a secondary center of diversity in the Andean region [18,21]. From tropical America, the cherimoya was introduced to continental Spain between the 16th and 18th centuries. From there, this fruit species was further introduced to Italy, Portugal, and northern Africa [22]. After that, it was grown in most tropical regions in Asia. Today, cherimoya is cultivated in tropical and subtropical regions throughout the world, including the Americas, Asia, Africa, Australia, and the Mediterranean region. Spain is the largest cherimoya-producing country, followed by Peru, Chile, Ecuador, and Mexico [21].

The *Magnoliales*, together with the Canellales, Laurales, and Piperales, form the Magnoliid complex [23]. The phylogenetic relationships among the clades have undergone intensive investigations in recent years, but the results have been conflicting. A recent study [24] analyzed four possible tree topologies by using different sampling datasets and favored the topology that monocots and magnoliids are closer to each other than to eudicots. Using single-copy orthologous gene trees, three other studies [25,26,27] reported that eudicots and monocots form a clade that is a sister to magnoliids. Other recent studies also showed that magnoliids and eudicots are closer to each other than they are to monocots [28,29,30]. It is expected that additional magnoliid genomes will help resolve the previous conflicting conclusions regarding the phylogenetic relationship of the three major angiosperm clades. Furthermore, new magnoliid genomes would improve our understanding of the phylogenetic position and divergence time of *A. cherimola* from other magnoliids. 

The economic importance of cherimoya and significance of *Annona* species in plant evolution warrant greatly expanded efforts in developing genomic tools for supporting the conservation of *Annona* genetic diversity, crop genetic improvement and utilization of *Annona* genetic resources in the food and pharmaceutical industries. However, progress in developing genomic resources for *Annona* lags behind what has been achieved for temperate fruit trees and annual crops. While the first sequenced genome in the *Annona* genus, *A. muricata*, was published in 2021, only raw DNA reads are available in GenBank, and its genome assembly is still unavailable. More recently, the first *A. cherimola* genome was developed from a Spanish cultivar, “Fino de Jete” [31].

In this study, we present the chromosome-level genome assembly of the American cherimoya cultivar “Booth”. Our objectives were to: (1) explore the intra-specific genomic diversity in the *A. cherimola* species; (2) analyze possible major duplication events in cherimoya; (3) assess phylogenetic relationships among monocots, eudicots, and magnoliids, (4) study disease-related genes and pathways, and (5) identify potential acetogenin biosynthetic gene clusters in the cherimoya genome.

## 2. Materials and Methods

### 2.1. Plant Materials

Three healthy young leaves of cherimoya cultivar ‘Booth’ (Appendix A) were collected from the greenhouse of USDA-ARS, Sustainable Perennial Crops Laboratory, Beltsville, Maryland. From the leaf tissue, DNA was extracted and used for generation of the reference genome. ‘Booth’ is one of the most planted cherimoya cultivars in the United States. This cultivar is known for its sweetness and broad adaptability under different environmental conditions [32]. 

### 2.2. PacBio Sequencing and Assembly

DNA samples were quantified using a Qubit 2.0 Fluorometer (Life Technologies, Carlsbad, CA, USA). The PacBio SMRTbell library (~20 kb) for PacBio Sequel was constructed using SMRTbell Express Template Prep Kit 2.0 (PacBio, Menlo Park, CA, USA) using the manufacturer’s recommended protocol. The library was bound to polymerase using the Sequel II Binding Kit 2.0 (PacBio) and loaded onto PacBio Sequel II). Sequencing was performed on PacBio Sequel II 8M SMRT cells generating 13.6 gigabases of data.

Next, 13.6 (21.8×) gigabases of PacBio Hifi reads were used as an input to Hifiasm1 v0.15.4-r347 with default parameters. Blast results of the Hifiasm output assembly (hifiasm.p_ctg.fa) against the nt database were used as input for blobtools2 v1.1.1 and scaffolds identified as possible contaminants were removed from the assembly (filtered.asm.cns.fa). Finally, purge_dups3 v1.2.5 was used to remove haplotigs and contig overlaps (purged.fa).

### 2.3. Dovetail Hi-C Sequencing and Assembly Scaffolding with HiRise

Dovetail Omni-C library was constructed as previously described [33,34]. The library was sequenced on an Illumina HiSeqX platform to produce an approximately 112.2× sequence coverage. Then, HiRise used MQ > 50 reads for scaffolding.

The *de novo* Hifiasm assembly and Dovetail Omni-C library reads were used as input data for HiRise (https://github.com/DovetailGenomics/HiRise_July2015_GR, accessed on 17 January 2024), a software pipeline designed specifically for using proximity ligation data to scaffold genome assemblies [35]. Dovetail Omni-C library sequences were aligned to the draft input assembly using BWA-0.7.17 (r1188) (https://github.com/lh3/bwa, accessed on 17 January 2024). The separations of Dovetail Omni-C read pairs mapped within draft scaffolds were analyzed by HiRise to produce a likelihood model for genomic distance between read pairs, and the model was used to identify and break putative misjoins, to score prospective joins, and make joins above a threshold.

### 2.4. Repeat and Noncoding RNA Annotation

The *de novo* repeat library was generated using RepeatModeler [36] (http://www.repeatmasker.org, accessed on 17 January 2024). LTR_retriever [37] and MITE tracker [38] with default parameters, then repetitive elements in the genome sequences were identified using RepeatMasker [39] and the repeat library. The predicted repeats with unknown types were classified using Teclass [40]. Transfer RNA (tRNA) genes were predicted using tRNAscan-SE [41] in eukaryotic mode, while ribosomal RNA (rRNA) genes were predicted by using Barrnap [42] with the eukaryotic database. Noncoding RNA genes were detected by using Infernal [43] against the Rfam database [44]. 

### 2.5. Protein-Coding Gene Prediction

The protein-coding genes were predicted using the approach described in our previous studies [33,34]. The ab initio and homology-based approaches were combined in the MAKER pipeline [45]. In the homology-based approach, the proteins from 12 species, *Aristolochia fimbriata* (Afi), *Arabidopsis thaliana* (Ath), *Amborella trichopoda* (Atr), *Cinnamomum kanehirae* (Cka), *Chimonanthus salicifolius* (Csa), *Liriodendron chinense* (Lch), *Litsea cubeba* (Lcu), *Magnolia officinalis* (Mof), *Nelumbo nucifera* (Nnu), *Oryza sativa* (Osa), *Persea americana* (Pam), *Piper nigrum* (Pni), and all proteins in the Swiss-Prot database were used as protein evidence. Three RNA-seq data for atemoya (SRR6031481, SRR6031482 and SRR5908896) were assembled by using Trinity [46] to obtain the transcripts and then served as the transcriptome evidence. The protein and transcriptome evidence was used in MAKER to guide the ab initio gene predictions. In the ab initio method, SNAP [47] and Augustus [48] were trained after each round of MAKER and the training results were integrated for the next round. MAKER was run with three iterations to ensure gene prediction accuracy. The final gene models were selected from the third-round output using protein lengths larger than 30 amino acids and then checked for annotation completeness using BUSCO (-l viridiplantae, m proteins) [49]. 

### 2.6. Orthologous Gene Clusters and Phylogenetic Analyses

Orthogroups were constructed with the *A. cherimola* genome and 19 other plant genomes—4 eudicots (*A. thaliana*, *Coffea canephora*, *N. nucifera*, *Vitis vinifera*); 4 monocots (*Musa acuminata*, *O. sativa*, *Phalaenopsis equestris*, *Sorghum bicolor*); 9 magnoliids (*A. fimbriata*, *C. kanehirae*, *C. salicifolius*, *L. chinense*, *L. cubeba*, *Magnolia biondii*, *M. officinalis*, *P. americana*, *P. nigrum*); 1 basal angiosperm (*A. trichopoda*) and 1 gymnosperm (*Ginkgo biloba*), using the software OrthoFinder v2.5.4 [50] with sequence search program Mmseqs [51]. All the genomes and their annotation files were downloaded from the links in their papers, the NCBI database, or provided by the authors. However, the genome and annotation files for *A. muricata* were not available (the authors were contacted but no responses were received). The low-copy orthogroups were selected if the number of orthologues ≤2 in each species. Each orthogroup was aligned with Muscle [52], and all alignments were integrated and trimmed using Gblocks [53]. Then, the species phylogenetic tree was constructed using IQ-TREE [54] with 1000 replicates for ultrafast bootstrap and automatic best-fit model selection using “m MFP” mode. 

To estimate the divergence time, the software r8s v1.8.1 [55] was used with Penalized Likelihood (PL) method and three fossil calibrations from TimeTree database [56]: (i) the divergent time of Gbi and Atr was ~330 million years ago (Mya); (ii) the divergent time of Ath and Nnu was between 119 and 125 Mya; (iii) the divergence time of Osa and Mac was between 98 and 116 Mya. The species tree with branch lengths generated by IQ-TREE was used as input. 

The gene families that had significant expansion and contraction were analyzed for each node across the species time tree using CAFE5 [57]. The large orthologous gene clusters (containing ≥ 100 orthologues in at least one species) were removed from analysis as suggested in CAFE5 pipeline. The species time tree generated by r8s was used as input. The significantly expanded or contracted gene families were determined based on *p*-values ≤ 0.05. 

### 2.7. GO Enrichment Analysis

All proteins of the 20 genomes were annotated by using emapper [58] searching eggNOG database with e-value < 10^−5^. The GO enrichment analysis was performed using the binomial test described in our previous analysis [33,34]. First, the foreground and background datasets were selected for different purposes (See Section 3). Second, the GO terms at different levels were converted to the 5th level using GOATOOLS [59], and the 5th-level GO terms in both datasets were counted as inputs. Next, the “binom_test” function in the Python package scipy [60] was applied to calculate the *p*-value for each GO term, and all *p*-values were adjusted to correct multiple testing using the “p.adjust” function in R. Lastly, the significantly enriched GO terms in the foreground were determined if the adjusted *p*-value ≤ 0.05. 

### 2.8. Genome Synteny

To study major duplication events in the *. cherimola* genome, synteny searches were performed to compare the genome structures of *A. cherimola* with those of the Atr and Afi genome using MCscanX [61]. WGDI [62] was used to identify self–self synteny for six species (Ach, Mof, Mbi, Csa, Cka and Lcu) that have chromosomal-level genome assembly. The syntenic blocks were selected if there were at least five gene pairs. The number of synonymous substitutions per synonymous site (Ks) was calculated for each gene pair in syntenic blocks as described in the MCscanX and WGDI pipeline. The plot for syntenic block comparison was generated using MCscan python version [63]. The dot plot of self-synteny in *A. cherimola* was generated using WGDI, and the enrichment of functions for all duplicated genes in syntenic blocks was studied. In addition to syntenic comparison, the Ks values were calculated with the whole paranome (all paralogous genes in a genome) of eight species in *Magnoliales* and Laurales orders using the WGD program [64]. 

### 2.9. Plant Resistance Gene Identification

Plant resistance genes were identified by using RGAugury, in which R genes were grouped into 11 classifications based on the presence of domains, including nucleotide-binding site (NBS), leucine rich repeat (LRR), transmembrane (TM), serine/threonine and tyrosine kinase (STTK), lysin motif (LysM), coiled-coil (CC) and Toll/Interleukin-1 receptor (TIR) [65]. We then focused on the NB-ARC domains genes involved in intracellular nucleotide-binding leucine-rich repeat receptors (NLRs) to detect effectors activating effector-triggered immunity. NBS domain sequences were identified by pfam_scan and then aligned by MAFFT [66]. Polygenetic tree of NBS domains was built using FastTree [67]. The tree was visualized with iToL [68].

### 2.10. Acetogenin (ACG) Biosynthetic Gene Cluster Identification

All potential plant secondary metabolite gene clusters were identified by using plantiSMASH [69] with a genome sequence and GFF file annotated by MAKER as input. The results only contained one polyketide biosynthetic gene cluster. All the protein sequences from the potential ACG gene cluster were searched against TAIR10 with E-value < 1 × 10^−6^, and the best hit was used to infer the function. 

## 3. Results

### 3.1. The Chromosome-Level Genome Assembly of Cherimoya “Booth”

The *A. cherimola* “Booth” genome was sequenced and assembled by Dovetail Genomics. First, a total of 13.6 Gb (21.8×) PacBio Hifi reads, and 70.1 Gb (112.2×) Illumina reads (Omni-C library) were generated. The genome size was estimated to be approximately 625 Mb based on Omni-C Illumina reads, and the heterozygosity rate was 1.14% (Appendix A) according to GenomeScope (http://qb.cshl.edu/genomescope/, accessed on 17 January 2024). Then, the “Booth” genome was assembled into 1658 contigs using Hifi reads. The contigs were further scaffolded into a chromosome-level assembly with the Omni-C reads using Dovetail HiRise™ scaffolding software (https://github.com/DovetailGenomics/HiRise_July2015_GR, accessed on 17 January 2024). In total, 1377 scaffolds (794 Mb) were in the final assembly with a scaffold N50 = 97.59 Mb (Table 1). A total of 284 gaps remained and the percentage of Ns was only 0.00359% in the assembly. Seven linkage groups were identified from the link density histogram (Figure 1A) corresponding to seven pseudo-chromosomes in the *A. cherimola* genome. This is consistent with the seven chromosomes in its sister species *A. muricata* [70] and agrees with the flow cytometry result [71]. The seven chromosomes also have telomere repeats in at least one of their two ends and centromere repeats (Appendix A). The seven chromosomes account for 87.6% of the total genome length; the chromosomes were ordered by their sizes, from the largest (~128 Mb) to the smallest (~73 Mb), and named from AC1 to AC7 (Figure 1B). Regarding the core gene completeness of the genome assembly, 98.43% (251/255) of complete core genes were found in the genome assembly in BUSCO (Benchmarking Universal Single-Copy Orthologs) analysis (Table 1). 

Notably, repeat sequences accounted for 68.23% of the genome, in which long terminal repeat (LTR) retrotransposons were the most abundant (25.49%) (Appendix A). The percentage of total repeats in *A. cherimola* was higher than its closely related species *A. muricata* (54.87%) [70], while the percentage of LTR retrotransposons in *A. cherimola* was lower than that in *A. muricata* (41.28%). The percentage of repeat sequences in *A. cherimola* was similar to that of two species in *Magnoliales* order, 66.48% in *M. biondii* [72] and 61.64% in *L. chinense* [73], but lower than 81.44% in *M. officinalis* [27]. Among the LTR retrotransposons, the number of Gypsy elements was larger than the number of Copia elements; however, the total length of Copia elements was longer. In addition, the unevenly distributed LTR retrotransposons in *A. cherimola* genomes tended to accumulate in sequence regions that have lower gene density, while LINE-1 elements showed the opposite trend (Figure 1B).

In *A. cherimola* “Booth” genome, 45,272 protein-coding genes (≥30 aa) were predicted with 92.9% of complete BUSCOs (protein mode) as the gene annotation completeness. The number of predicted genes in *A. cherimola* lowered to 34,890 if filtered with ≥100 aa sequence length; the gene number in *A. cherimola* was thus much higher than that in *A. muricata* (23,375 genes) (Table 1). A total of 32,377 genes (71.52% of 45,272 genes) can be predicted with GO annotations, KEGG pathway, or Pfam annotation. In addition, a total of 495 transfer RNA (tRNA) and 61 ribosomal RNA (rRNA) genes were predicted in the seven chromosomes, while 1448 tRNA and 11,401 rRNA genes were predicted in the remaining 1370 scaffolds. This huge number of rRNA genes was unusual, and most of them were 5S ribosomal RNAs. Examination of the genomic locations of these rRNA genes found that only 61 genes were located on the seven chromosomes and the rest were on unplaced scaffolds (Appendix A). This indicates a possibility of mis-annotation of rRNA genes, although it is not uncommon that plants may contain a large copy number of rRNA genes [74].

### 3.2. No Clear Whole Genome Duplication Event Was Found in Cherimoya

Genome polyploidy, through whole genome duplication (WGD), plays an important role in plant evolution [75]. Commonly, WGD analysis is performed by detecting colinear syntenic blocks and calculating the paralogous gene synonymous substitution rate (Ks). *A. trichopoda* (Atr) is known to lack any lineage-specific WGDs except for the ancient WGD shared by all flowering plants. Therefore, newly sequenced genomes are usually compared with *A. trichopoda* to identify whether there are lineage-specific WGDs [24,76]. We first identified all the syntenic blocks in *A. cherimola* “Booth” genome and then plotted the Ks density distribution of all paralogs in the syntenic blocks. We found that *A. cherimola* (Ach, green curve in Figure 2A) has a clear intra-genome Ks peak at ~1.1. The peak is smaller than the inter-genome syntenic block Ks peaks for syntenic ortholog comparisons between Ach and the two references (Ks~1.25 for Ach-Atr and Ach-Afi, Afi for *A. fimbriata*) and between the two references (Ks~1.5 for Afi-Atr) (Figure 2A). 

However, when plotting the all the paralogs in the genomes (i.e., whole paranome), we observed a flat Ks~1.35 peak in *A. cherimola* (Figure 2B, see below). For comparison with other magnoliid genomes, we have also plotted the whole paranome Ks of *L. chinense*, *M. officinalis*, and *M. biondii* of the *Magnoliales* order, which have a Ks peak at 0.7–0.8. This peak at 0.7–0.8 corresponds to a recent WGD event and is consistent with results determined in previous studies for the three species [27,72,73]. Obviously, *A. cherimola* does not share this WGD event as it does not have this Ks peak at 0.7–0.8.

The inter-chromosome syntenic block collinearity dot plot also confirms the lack of a recent WGD event, as the inter-chromosome syntenic blocks are all short and fragmented (Figure 1B and Figure 2C). Instead of WGD, these short blocks might be a result of segmental duplications, which are smaller-scale duplications than WGD. 

### 3.3. Magnoliids Are Phylogenetically Closer to Eudicots Than to Monocots

As introduced in the Introduction, the phylogenetic closeness among the three major clades within flowering plants has been under debate for years. Our phylogenetic tree supported that magnoliid and eudicot lineages shared more recent common ancestry compared to monocots (Figure 3A), which also has been suggested in other studies [28,29,30]. In our phylogenetic tree reconstruction, we included genomes of 10 magnoliids, 4 monocots and 4 dicots, with *A. trichopoda* and *Ginkgo biloba* as the outgroups. In total 852 low-copy orthologous genes were used for sequence alignment and phylogenetic analysis. The bootstrap values in most nodes were 100, indicating high confidence in the tree topology (Figure 3A). Using this tree topology as input, the divergence times of each node in the tree were estimated by the software r8s (see Section 2). Monocots were inferred to have separated from magnoliids and eudicots 180 million years ago (Mya), and then magnoliids and eudicots diverged 170 Mya. 

Within the magnoliid lineage, the phylogenetic relationships among Piperales, *Magnoliales* and Laurales orders were highly consistent with previous findings [72]. Piperales diverged from the other two orders 158 Mya, while the divergence of *Magnoliales* and Laurales happened 124 Mya. In addition, *A. cherimola* separated from other *Magnoliales* species 84 Mya. Unfortunately, the *A. muricata* genome and genes are not publicly available, which hindered an inference of the divergence of the two *Annona* species. 

Based on the analysis of gene family expansion and contraction along the species tree, a total of 94 gene families have been significantly expanded in *A. cherimola* “Booth”, while 51 gene families significantly contracted (Figure 3A). A GO enrichment analysis was conducted with the significantly expanded/contracted gene families used as the foreground, and the gene families shared among three major angiosperm clades (gene orthologues found in at least one species in each clade) as the background. The results showed that genes related to terpene and the terpenoid metabolic process (GO:0006721, GO:0042214) or biosynthetic process (GO:0010333, GO:0046246) were significantly expanded in the *A. cherimola* genome (Figure 3B); in contrast, genes related to positive regulations in cellular activities (e.g., stress response and signaling) were significantly contracted (Appendix A). Note that the GO annotation was based on a sequence similarity search against the eggNOG database, and should not be used as a direct evidence to claim the existence of specific metabolites. For example, the glucosinolate biosynthesis process (GO:0019758) is found in Figure 3B, but the presence of glucosinolates in *Annonaceae* has never been confirmed.

### 3.4. Comparative Genomics Identifies Intra-Specific Genomic Variations within Annona cherimola

To determine the genomic variations within the *Annona* genus, we have compared the *A. cherimola* “Booth” genome with the “Fino de Jete” genome. *A. cherimola* cultivar “Fino de Jete” is grown in Spain and its genome was published in February 2023 [31]. The Spanish “Fino de Jete” chromosome-level genome assembly has a size of 1.13 Gb, which is 343 Mb larger than the American “Booth” genome assembly (Table 1). The “Fino de Jete” genome was reported to contain 743.28 Mb repeats [31], while our “Booth” contains 539.92 Mb repeats. Therefore, the “Fino de Jete” genome has 201.54 Mb extra repeats than the “Booth” genome, accounting for 58.9% of the size difference of the two genomes. To compare the two assemblies at the genomic DNA level, we used MUMmer [77] to generate a whole genome alignment (Figure 4A). Despite the large difference in genome size, the two genomes share >99% nucleotide sequence identity and very similar chromosomal co-linearity, except for some major structural variations in Chr4 and Chr6. All chromosomes except for Chr4 are longer in “Fino de Jete” than in “Booth”. The protein-coding gene contents of the two genomes were also compared with MMSeqs2 [78], which revealed that 41,080 (99.2%) “Fino de Jete” genes and 34,827 (76.2%) “Booth” genes are shared between the two genomes. 

To study when the two cultivars diverged, we built a species phylogeny using 1330 single-copy orthogroups that are shared by five genomes of the *Magnoliales* order and *Arabidopsis*. The species phylogeny was used to date the divergence times, using three fossil calibrations from the TimeTree database [56]: (i) the divergent time of Ath and Lch was ~163 million years ago (Mya), (ii) the divergent time of Lch and Ach was between 95 and 113 Mya, and (iii) the divergence time of Lch and Mof was between 28 and 50 Mya. These results are consistent with what is shown in Figure 3A. The two cherimoya cultivars, “Booth” and “Fino de Jete”, were estimated to have diverged from each other about 1.93 Mya (Figure 4B). These two cultivars were brought to Spain and the United States only a few hundred years ago. The divergence time estimation result demonstrated their different origin, and different domestication before they were imported from the Neotropical region.

### 3.5. Terpene/Terpenoid Metabolism Functions Are Enriched in Magnoliales

Proteins of all 20 genomes of Figure 3A were used to build orthologous gene clusters (OGCs). Comparing OGCs shared among the three major clades in angiosperms revealed that 10,396 OGCs were conserved in all three clades (Figure 5A). In comparison, 11,698 OGCs only contained species from the magnoliids clade. Among the 11,698 OGCs unique to the magnoliids, 276 OGCs were shared by all three orders (Figure 5B), and 2863 OGCs only contained species in the *Magnoliales* order (Figure 5A,B). Magnoliids shared more OGCS with eudicots (2316, Figure 5A) than with monocots (533), supporting a closer relationship between magnoliids and eudicots (Figure 3A). The closer relationship between *Magnoliales* and Laurales orders is also supported by the higher number of shared OGCs (762, Figure 5B). 

To study which GO functions are enriched in genes unique to the magnoliid clade, binomial tests were performed for a GO enrichment analysis (see Section 2). Specifically, the 11,698 OGCs unique to the magnoliid clade were used as the foreground, while the 10,396 core OGCs (Figure 5A) were used as the background. Similarly, GO enrichment analysis was also performed to identify enriched functions in the *Magnoliales* order; 2863 OGCs unique to *Magnoliales* order were used as the foreground, and 10,396 core OGCs used as the background. Species in the *Magnoliales* order or magnoliids clade had enriched functions related to terpene synthesis or terpenoid metabolism from the GO enrichment analysis results (Figure 5C and Appendix A). Terpenes are natural products found in plants and are responsible for their fragrance, taste, and pigment [79]. Terpenoids are responsible for plants’ defense against biotic and abiotic stresses, used as signal molecules to attract insects for pollination, and have substantial pharmacological bioactivity [80]. The biosynthesis of terpenoids has been well studied in many magnoliid species, including *M. biondii* [72], *L. cubeba* [29], *A. fimbriata* [24] and *C. kanehirae* [28]. 

### 3.6. Plant Resistance TNL Genes Are Absent in Magnoliales

Plant disease resistance proteins (R proteins) [81] are extremely important for plant immunity and protect plants from diseases. Therefore, the study of R proteins in *A. cherimola* will contribute to the understanding of cherimoya disease resistance for increased production. Most R proteins contain the nucleotide-binding site (NBS) domain and the Leu-rich repeat (LRR) domain and are commonly known as NBS-LRR or NLR proteins. According to the presence of other functional domains, R proteins are classified into four main classes: (1) TNLs (**T**oll/Interleukin-1-**N**BS-**L**RR), (2) CNLs (**C**oiled-coil (CC)-**N**BS-**L**RR), (3) RLKs (**r**eceptor-**l**ike protein **k**inases), and (4) RLPs (**r**eceptor-**l**ike **p**roteins). TNLs and CNLs are R proteins with NBS domains, while RLKs and RLPs do not have NBS domains but TM (transmembrane) domains. Previous studies have shown that TNL are rarely found in monocots [82,83] while both TNL and CNL are present in basal angiosperms [84].

Using RGAugury [65], we identified in total 439 NLR (NBS-LRR) proteins in the *A. cherimola* genome (Table 2). Interestingly, no TNLs are found in genomes of the *Magnoliales* order (Ach, Mof, Mbi and Lch) and the Piperales order (Afi, Pni). The four genomes of the Laurales order contain between 0 and 3 TNL genes. Therefore, like monocots, magnoliids tend to have no or very few TNL genes. Monocots and magnoliids also have much fewer TN and TX genes than dicots, suggesting a low abundance of Toll/Interleukin-1 domains. Considering the non-angiosperms (Amborella and Ginkgo) have a higher number of TNL, TN, and TX genes, independent gene loss can be used to explain the low abundance of these genes in monocots and magnoliids. To study the evolution of NLR proteins, we built a phylogenetic tree using all the NBS domain sequences from five species (Ach, Lcu, Ath, Osa and Gbi, shown with different colors in the circular ring in Figure 6). The NBS domains of TNLs/TNs (yellow and purple branches) are predominantly found in Ath (green in the ring) and Gbi (cyan in the ring), and are separated from the NBS domains of CNL/CN/NL proteins. Lcu of *Magnoliales* (red in the ring) does contain a small number of sequences (Table 2) clustered within the TNL clade (Figure 6). The NBS domains of CNL/CN/NL proteins were further classified into subclades. Gbi (cyan in the ring) is much more narrowly distributed than other species, which suggests a major expansion of NLRs in flowering plants.

### 3.7. A Potential Acetogenin Biosynthetic Gene Cluster in A. cherimola Genome

Acetogenins (ACGs), specifically the tetrahydrofuran (THF) acetogenins, are a class of lipophilic polyketide natural products consisting of C32~C37 long-chain fatty acids and intermittent THF rings and a terminal lactone ring [16,85] (Figure 7A). THF ACGs are uniquely found in the family *Annonaceae* and have great potential for treating cancers due to their ability to inhibit tumor cell proliferation [13]. ACGs are found in the roots, seeds, pulp and leaves of cherimoya (16). To discover the candidate gene cluster responsible for the synthesis of ACGs, plantiSMASH [69] was run on the *A. cherimola* “Booth” genome. A total of 24 biosynthetic gene clusters (BGCs, Appendix A) were predicted, including 1 polyketide BGC. This polyketide BGC is inferred to be the most likely candidate gene cluster for the synthesis of ACGs in *A. cherimola* for the following reasons. 

This ACG-BGC contains nine genes (Ach20829-Ach20837, total 97 kb, Figure 7B) according to plantiSMASH, encoding proteins with Pfam domains annotated for polyketide synthesis. Ach20829 contains K-box and SRF-TF domains, and its best *A. thaliana* hit (AT4G11880) is a MADS-box protein, which might control the expression of the ACG-BGC. Ach20830, Ach20831, Ach20833, and Ach20834 are four proteins that contain an N-terminal FAE1_CUT1_RppA domain and a C-terminal ACP_syn_III_C domain. Their best *A. thaliana* hits (AT1G04220 and AT2G26640) are members of the 3-ketoacyl-CoA synthase family for the synthesis of VLCFAs (very long-chain fatty acids). Importantly, according to the plantiSMASH signature gene search result of this BGC, Ach20830, Ach20831, and Ach20834 are all chalcone synthases and contain a Chal_sti_synt_C domain, which may be responsible for adding the lactone ring in ACGs. Ach20832 contains two domains (Copine and zf-C3HC4_3), and its best hit (AT3G01650) is an E3 ubiquitin–protein ligase. Ach20835 has its best *A. thaliana* hit (AT3G22990) annotated as a SWI/SNF chromatin-remodeling complex (CRC) component LFR (leaf- and flower-related) protein. Ach20836 has an Epimerase domain, and its best *A. thaliana* hit (AT5G28840) is a GDP-mannose 3,5-epimerase; plantiSMASH also annotates it as the signature gene of this BGC. The last gene in the BGC Ach20837 has a PDDEXK_6 domain, formerly known as DUF506, and its best *A. thaliana* hit (AT3G22970) were found to inhibit root hair elongation [86].

## 4. Discussion

Cherimoya is a commercially important crop known for its delicious fruits and valuable bioactive compounds. It is thought to have originated from the Andes and Central America. Over a dozen cherimoya cultivars are described in the literature [87]. In this study, we selected a Californian cultivar “Booth” [88] for genome sequencing. The first cherimoya genome was published in 2023 from a Spanish cherimoya cultivar “Fino de Jete”. Both “Booth” and “Fino de Jete” genomes were assembled into a chromosome-level assembly at a similar level of genome quality (Table 1), which allows a whole genome alignment analysis (Figure 4A and Appendix A). It is striking that the two genomes have a significant difference in genome size (343 Mb longer in “Fino de Jete”, Table 1), which is at least partially due to the longer repeat regions in the “Fino de Jete” genome. Despite the genome size difference, the two genomes show quite similar chromosome co-linearity and high nucleotide sequence identity (Figure 4A). Major chromosomal rearrangements are observed between “Booth” Chr6 and “Fino de Jete” Chr2. Interestingly, “Booth” Chr6 has telomere repeats identified at both ends (Appendix A), suggesting this major structural inversion might demonstrate a real difference between the two closely related genomes. Most “Fino de Jete” genes are also present in “Booth” but not the other way around. Searching the 10,875 “Booth” genome-specific genes against the “Fino de Jete” genome using BLAT [89] found that 70% of them are present in the “Fino de Jete” genome and 48% of them encode short proteins (<100 aa) (Appendix A). Almost 80% of these “Booth” genome-specific genes do not have *Arabidopsis* homologs, and when they do, most match *Arabidopsis* proteins encoded or located in chloroplast. Obviously, the gene models predicted in both genomes are not perfect and will need future improvement.

The high heterozygosity rate is observed in both “Booth” (1.14%) and “Fino de Jete” (1.05%) genomes, suggesting that high-quality phased haploid genomes will be needed in the future. A newer version of Hifiasm (v0.19.5-r593) introduced a purge function that can assemble haploid genomes. We have tried this new version using the 13.6 G Hifi reads and obtained two phased genomes at 749.5 M and 700.5 M, which are close to the draft genome size (794 M) reported in this paper. Additional sequencing will be needed in the future to generate the high-quality phased haploid genomes. With the current draft assembly, the 98.43% complete BUSCOs of the “Booth” genome indicates the draft genome has a great coverage of the protein coding genes, which is appropriate to perform the comparative genomics analyses in this paper.

A genome divergence time estimation based on a sequence alignment of 1330 single-copy genes shows that the two cultivars “Booth” and “Fino de Jete” diverged from each other 1.93 Mya. This result suggests that the original plants of the two cultivars were of different origins, and possibly with different domestication histories, too, when they were brought to North America and Europe from the Neotropics a few hundred years ago. The divergence of these populations started in the early Pleistocene (2.58–0.773 Ma), which is consistent with the divergence time of many other plants in the Neotropical region [90,91]. The observed intraspecific genetic differentiation is also compatible with a recent finding [92], which reported two different haplotypes in the germplasm collected from three Central American countries (Honduras, Guatemala and Costa Rica). However, based on microsatellite analysis of cherimoya germplasm from tropical Americas, recent studies [18,21] also proposed that cherimoya originated in the highlands of Mesoamerica, and humans brought cherimoya from Mesoamerica to present-day Peru through long-distance sea-trade routes across the Pacific Ocean. Although the exact origin of the two cultivars (“Fino de Jete” and “Booth”) is unknown, our result shows that the hypothesis of the Mesoamerica—Andes dispersal of Cherimoya can be tested using comparative genomics between landraces from Mesoamerica and the Andes. In addition, morphological characteristics between these two cultivars should also be compared in future studies. 

The speciation of *A. cherimola* occurred around 84 Mya (Figure 3A), which is consistent with the estimated divergence time for *A. muricata* [72]. Based on 852 orthogroups for 20 species, the magnoliid clade has been shown to be closer to eudicots than monocots (Figure 3A), which has been reported in some studies [28,30,72]. Given that more species from magnoliid clade have been included than previous papers, our phylogenetic tree should be more accurate in terms of revealing the phylogenetic relationships among monocots, eudicots and magnoliids. 

Focusing on the orthologous gene clusters (OGCs) unique to the *Magnoliales* order, functions related to fruit flavors and plant stress responses, e.g., terpene synthesis or terpenoid metabolism are significantly enriched (Figure 5). This also agrees with the finding that terpene and terpenoid metabolic processes were significantly expanded in the *A. cherimola* genome (Figure 3B). Interestingly, the TNL genes are completely absent in genomes of the *Magnoliales* order and the Piperales order (Table 2), and the entire magnoliid clade has no or very few TNL genes, while the CNL sequence diversity is higher in magnoliids than in dicots and monocots (Figure 6).

Another major finding of this study is the candidate biosynthetic gene cluster (BGC) for THF acetogenin (ACG), which is the hallmark of the *Annonaceae* family with demonstrated anti-tumor activities. This was made possible by the *A. cherimola* “Booth” genome, because the BGC genome mining tool plantiSMASH has to use the assembled genome as input, which is the reason that the ACG BGC has never been identified before. Although experimental characterization is needed to confirm this BGC is indeed responsible for ACG synthesis, our sequence analysis of the nine member genes in this BGC strongly suggests it is the most likely candidate (Figure 7). First, plantiSMASH only found 1 polyketide BGC out of twenty-four total BGCs, and ACGs are lipophilic polyketide natural products. Second, four of the nine genes of the BGC are members of the 3-ketoacyl-CoA synthase family for the synthesis of VLCFAs (very long-chain fatty acids) according to their best *A. thaliana* hits and Pfam domains. Third, three of the 3-ketoacyl-CoA synthase genes might encode proteins with a Chal_sti_synt_C domain, which may be responsible for adding the lactone ring in ACGs. This BGC also contains other genes that may be important for regulating the ACG synthesis, such as the MAD-box protein and the SWI/SNF chromatin-remodeling complex (CRC) component LFR protein. Future experimental validation will be necessary to verify this BGC for acetogenin biosynthesis.

## 5. Conclusions

In summary, the cherimoya “Booth” genome, the second publicly available genome of the *Annonaceae* family, will be a valuable resource for studying the genetic diversity of *Annonaceae*, the evolution of magnoliids and flowering plants, the discovery of acetogenin biosynthetic genes and the origin, domestication and dispersal of cherimoya. It also provides novel genomic resources to support crop germplasm evaluation and new breeding strategies to improve the production of this economically important fruit crop.

## Figures and Tables

**Figure 1 plants-13-00636-f001:**
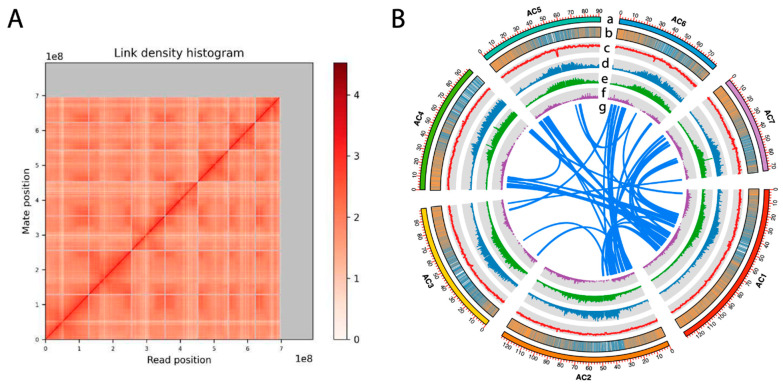
The chromosome-level genome assembly of cherimoya “Booth”. (**A**) Dovetail Genomic’s Hi-C linkage density heatmap for *A. cherimola* genome showing seven chromosomes. The darker color indicates a higher frequency of interaction. (**B**) The circos plot of the seven chromosomes. From outside to inside, we show (a) seven chromosomes arranged by size, (b) gene density (the number of genes found in each 200 kb sequence window, high and low gene densities are indicated in orange and blue, respectively), (c) GC content, (d) LTR Gypsy density, (e) LTR Copia density, (f) LINE-1 element density and (g) syntenic blocks between chromosomes.

**Figure 2 plants-13-00636-f002:**
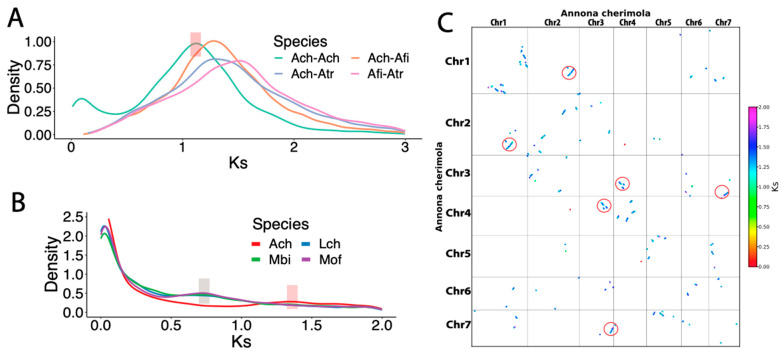
Duplication analysis of the *A. cherimola* genome. (**A**) The Ks distributions for all gene pairs in syntenic blocks computed by MCScanX. The three-letter codes for species are the 1st letter of genus + the 1st two letters of species (see Figure 3 for species name). The shades indicate the peaks. (**B**) The Ks distributions for the whole paranome of four species in *Magnoliales* order. (**C**) Dot plots of paralogs indicated self-syntenic blocks between chromosomes in the *A. cherimola* “Booth” genome. Dots are colored according to the Ks values.

**Figure 3 plants-13-00636-f003:**
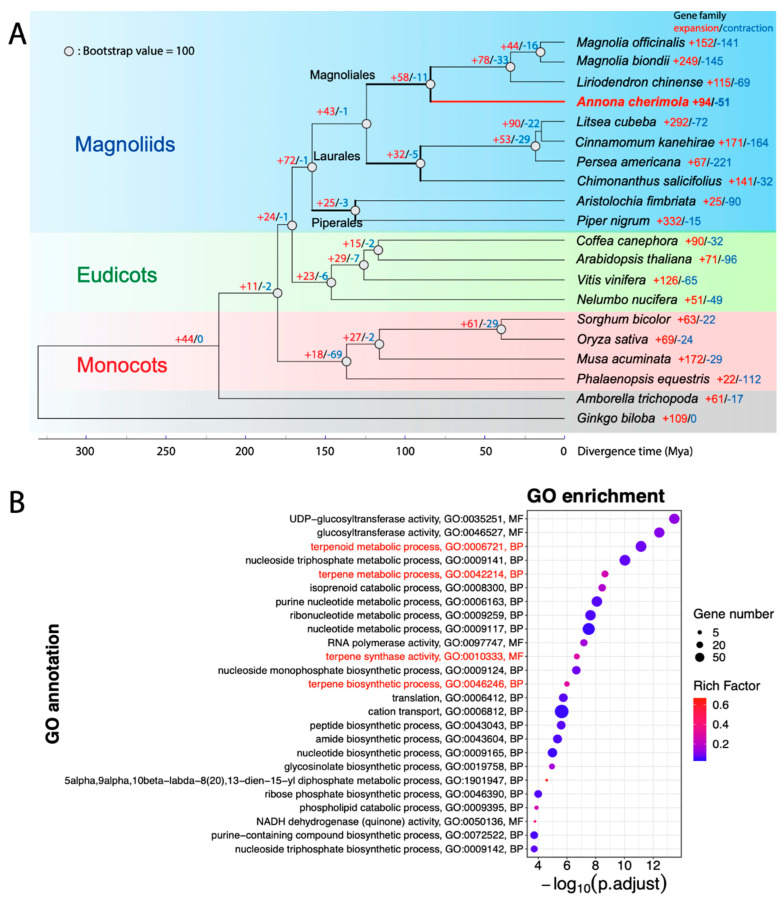
Phylogenetic relationships among the three major angiosperm clades and significantly expanded/contracted gene families along clades. (**A**) The phylogenetic tree with estimated divergence times for 20 species. A total of 18 species are selected from magnoliids (10 species), eudicots (4), monocots (4) and compared with *Amborella* and *Ginkgo* as outgroups. The bootstrap values equal to 100 are shown as gray circles on each node. The numbers of gene families significantly (*p* ≤ 0.05) expanded and contracted on each node are labeled in red (+) and blue (−), respectively. (**B**) The GO enrichment for 94 significantly expanded gene families in *A. cherimola*. The rich factor is the ratio of genes in significantly expanded gene families annotated with this GO function to all genes in the background (gene families shared by three major clades) annotated with this GO function. The terpene/terpenoids synthesis and metabolism functions are shown in red.

**Figure 4 plants-13-00636-f004:**
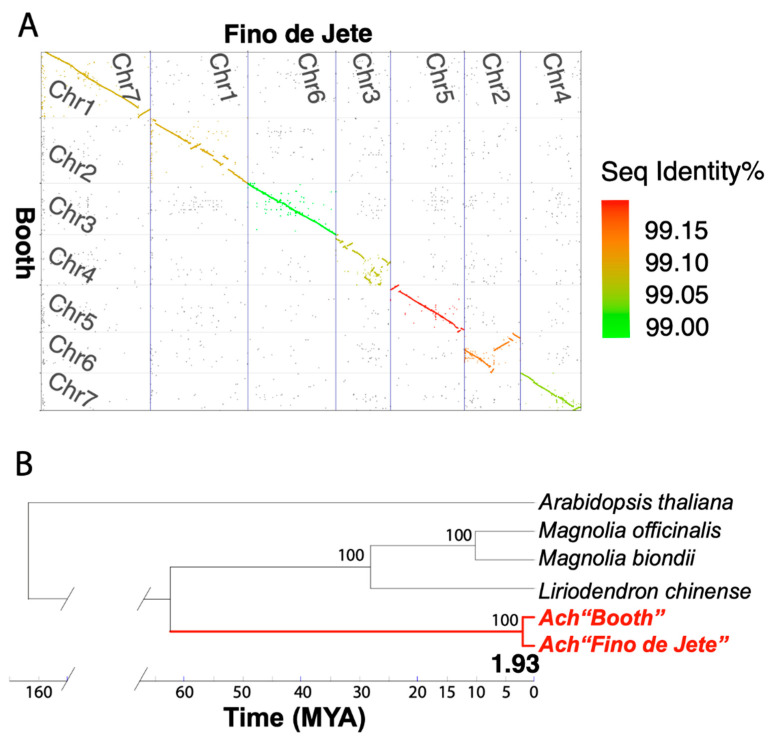
Genomic variations detected between two cherimoya genomes (“Fino de Jete” and “Booth”). (**A**) The dotplot of whole genome alignment of “Fino de Jete” and “Booth” genomes computed by MUMmer. Only the seven chromosomes of the two genome assemblies were aligned. The “Booth” chromosomes (*y*-axis) are numbered according to the Chr length, while the “Fino de Jete” chromosomes (*x*-axis) are not. Each dot in the plot is a fragment alignment. The color represents the average nucleotide sequence identity of all fragment alignments of each Chr. The Chrs are drawn in proportion to their lengths. (**B**) Phylogenetic relationship within the *Magnoliales* order. The bootstrap values equal to 100 are shown. The phylogenetic tree was constructed using 1330 single-copy orthogroups.

**Figure 5 plants-13-00636-f005:**
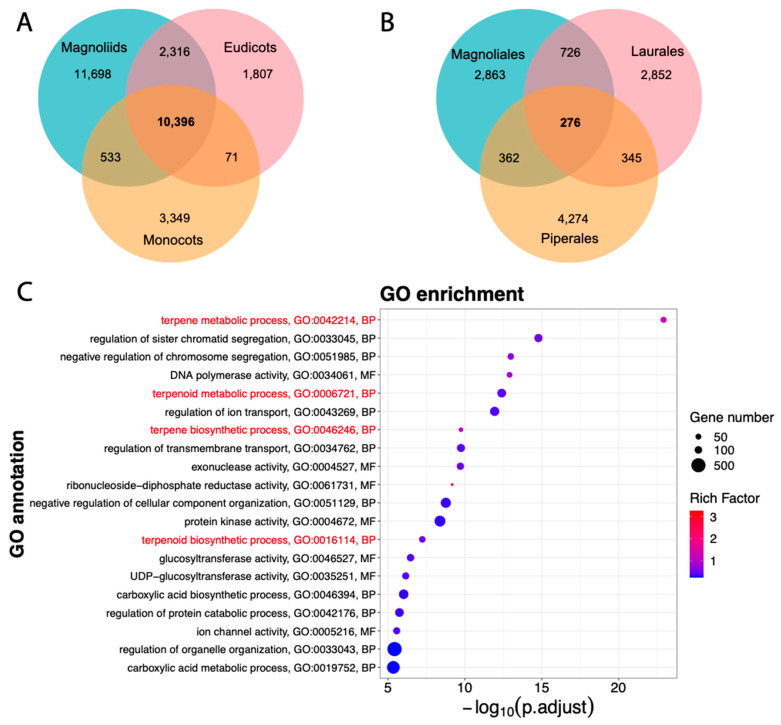
Orthologous gene clusters (OGCs) shared by and unique to different groups of genomes. (**A**) The Venn diagram shows that 10,396 OGCs are shared among three major clades of angiosperms. (**B**) The Venn diagram shows that 276 OGCs are shared among three major orders within the magnoliid clade. (**C**) The top 20 GO functions enriched in the 2863 unique OGCs that are only present in the *Magnoliales* order. The terpene/terpenoids synthesis and metabolism functions are shown in red. See Figure 3B for legends.

**Figure 6 plants-13-00636-f006:**
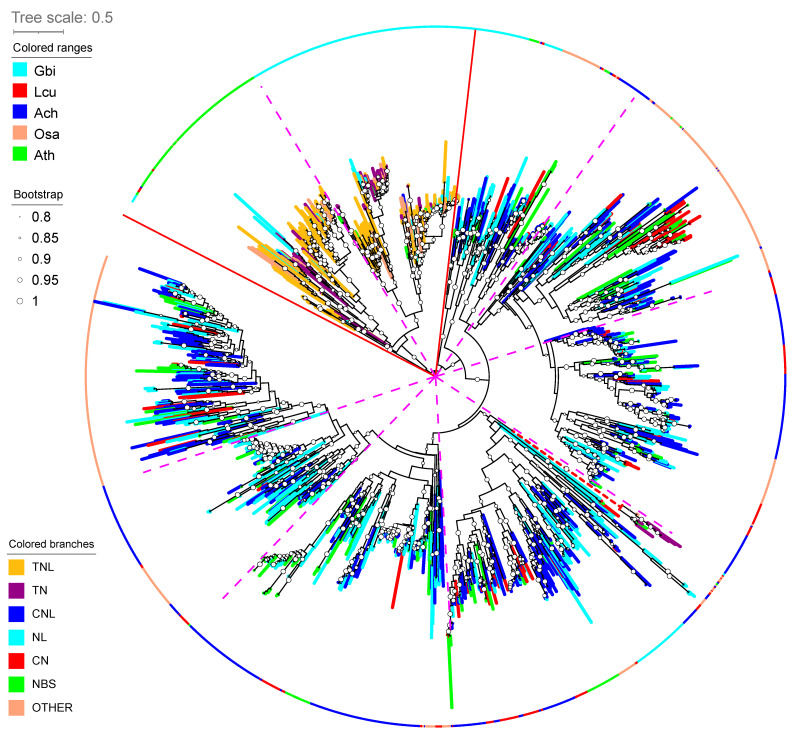
Phylogenetic tree of NBS domains from five representative plant genomes. The R gene family is colored in branches, and species is colored in the ring outside the tree. See Table 2 caption for the R gene family name explanation. The species’ three-letter codes are the 1st letter of genus + the 1st two letters of species. All the full species names can be found in Figure 3. Two red solid lines separate the whole tree into the TNL and CNL major clades. The TNL clade (TNL/TN) contains two sub-clades, TNL-Ath and TNL-Gbi. The CNL clade (all except for TNL/TN) can be divided into seven sub-clades based on the tree topology and presence/absence of the 5 species (pink dashed line). Some sub-clade contains NBS domains from all 5 species, while others contain fewer or single species.

**Figure 7 plants-13-00636-f007:**
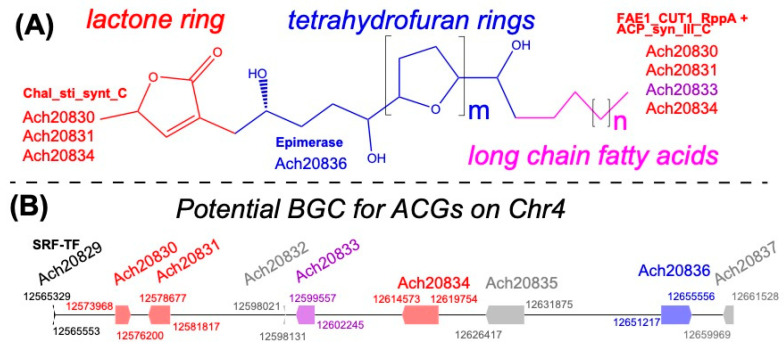
The putative Acetogenin (ACG) biosynthetic gene cluster. (**A**) Potential BGC1 (Ach20829-Ach20837) was predicted to synthesize polyketide by plantiSMASH. The domains of four signature genes are colored differently. (**B**) Overview of annonaceous ACG structure. ACG is composed of three parts: Lactone ring (red), Tetrahydrofuran rings (blue) and long-chain fatty acids (purple). The number of tetrahydrofuran rings varies from 1 to 3 (1 ≤ m ≤ 3). The Ach genes are colored differently corresponding to functions involved in three biosynthetic parts of ACG. Gray genes have functions not clearly relevant to ACG synthesis.

**Table 1 plants-13-00636-t001:** General statistics of assembly and annotations for cherimoya and soursop genomes.

Assembly	*A. cherimola* “Booth”	*A. cherimola* “Fino de Jete” *	*A. muricata* *
Total length (bp)	794,023,491	1,137,394,475	656,813,740
Number of scaffolds	1377	2052	755
Number of Chromosomes	7	7	7
Longest scaffold (bp)	128,576,476	212,253,197	122,620,176
Scaffolds N50 (bp)	97,591,913	170,859,109	93,205,713
GC content %	35.25	34.69	40.07
Complete BUSCOs %	98.43	93.0	-
**Annotation**			
Repeat sequences %	68.23	64.96	54.87
Number of protein-coding genes	45,272 (≥30 aa) ^#^	41,413 (≥50 aa)	23,375 (≥100 aa)
Number of genes with annotation	32,377	-	22,769
Complete BUSCOs %	92.9	90.9	92.14

* Statistics data were taken from soursop genome paper [70] and cherimoya “Fino de Jete” genome paper [31]. ^#^ 34,890 (≥100 aa), 43,926 (≥50 aa). BUSCO database: viridiplantae.

**Table 2 plants-13-00636-t002:** Distribution of NLR genes in 20 genomes.

Species	CNL	CN	NL	TNL	TN	TX	NBS	Other
Mof	125	28	97	0	1	6	18	0
Mbi	124	40	113	0	1	4	66	0
Lch	93	67	90	0	3	9	68	2
Ach	183	22	171	0	2	4	57	0
Lcu	64	20	51	3	5	9	29	0
Cka	174	32	128	2	3	9	31	0
Pam	6	4	22	1	0	6	13	0
Csa	81	23	53	0	1	6	25	4
Afi	15	1	12	0	2	3	2	0
Pni	115	43	168	0	7	2	203	0
Cca	267	43	330	3	2	4	76	2
Ath	39	1	21	75	14	34	5	15
Vvi	213	50	120	71	15	27	72	14
Nnu	71	11	40	21	3	6	16	5
Sbi	110	17	115	0	3	1	26	0
Osa	190	44	143	0	2	3	51	0
Mac	51	5	30	0	3	5	12	0
Peq	17	2	20	0	2	3	9	0
Atr	17	16	37	13	7	7	36	3
Gbi	33	2	67	91	36	16	9	20

NBS-encoding genes identified by RGAugury. The columns are the R gene family, named according to their domain compositions. C stands for coiled-coil domain; N stands for NBS domain; T stands for toll/Interleukin-1 domain; L stands for LRR domain; X stands for unknown domain; NBS includes proteins-only containing NBS domain; other type stands for chimeric domain/motif (with both C and T). See [65] for a detailed explanation of the R gene family names. The rows are species: the three-letter codes are the first letter of the genus + the first two letters of the species. For example, Mof is *Magnolia officinalis*. All the full species names can be found in Figure 3. The background colors indicate different groups of species and match the colors in Figure 3A.

## Data Availability

The *A. cherimola* “Booth” genome and gene annotation have been deposited in GenBank with a BioProject ID PRJNA954757 and are also made available at https://bcb.unl.edu/Ach/ (accessed on 17 January 2024).

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
