# Peer review of "The American Cherimoya Genome Reveals Insights into the Intra-Specific Divergence, the Evolution of Magnoliales, and a Putative Gene Cluster for Acetogenin Biosynthesis"

_plants, 2024, doi:10.3390/plants13050636_

Round 1

Reviewer 1 Report

Comments and Suggestions for Authors

Lines 70, 94, 97, 98, 609, 612

Although not clearly indicated so in the Instructions for Authors, authors' names should rather not be part of references in the text. Otherwise, the prescribed ACS format (using numbers instead of author names and years for references) would make close to no sense at all. There is also no obvious reason why authors are mentioned in a very limited fraction of in-text references only.

In an English text, "could" is ambiguous (was or would it be possible?) and should be replaced by clearer substitutes:

Line 100: "It is expected that additional magnoliid genomes *will* help (...)"

Line 109/110: "Moreover, more sequenced magnoliid genomes *can/may* help (...)"

Line 615/616: "our result shows that the hypothesis of Mesoamerica – Andes dispersal of Cherimoya *can* be tested (...)"

The sentence in lines 109-111 repeats text from lines 100-101 and should be deleted.

Lines 197-210: Italics (or abbreviations as introduced in the preceding paragraph) should be used for species names (as done in the rest of the text).

Line 250: "Magnoliales and Laurales order*s*"

Line 258: "focus*ed*"

Line 268: "contain*ed*"

Line 269: "search*ed*"

Line 351: "magnoliid" is printed in a different type for no obvious reason.

Line 365-366: Several switches between different types.

Lines 370-372: "approximately 180" and "around 170" should be explained by figures indicating the uncertainty and/or error bars in the time axis of Figure 3A

Line 377: italic type for genus names

Line 380: "enri*chment"; italic type for "A. cherimola"

Figure 3B: "glycosinolate biosynthetic process" (ca. 20 genes according to circle diameter) makes no sense in _A. cherimola_. Glucosinolates are narrowly confined to Brassicales/Capparales (eudicots), and none have reliably been detected in Annonaceae so far. Does "glycosinolate" include cyanogenic glycosides (that are biosynthetically related to glucosinolates but far more widespread) in the GO annotation? If this is the case, it should be explained in the methods or at least in the context of fig. 3B.

Line 382: add "The terpene/terpenoids synthesis and metabolism functions are shown in red."

Line 388: delete "occurred"

Line 422: italic type for "Arabidopsis"

Line 452: "*M*agnoliales and *L*aurales"

Line 459: "*M*agnoliales"

Lines 465-467: "*M*agnoliales"

Line 469: Not all terpenes (isoprenoids) are "aromatic compounds" in the sense of organic chemistry, not even are all of them cyclic. From their biosynthetic origin, all terpenes are (originally) branched and desaturated, but almost all features may be removed, altered or rearranged by subsequent modifications. There are even two fundamentally different routes leading to the common precursor (and the only truly common feature) of all isoprenoids, isopentenyl diphosphate.

Line 489: "Interesting*ly*"

Line 491: "Laurels" is probably an autocorrection glitch and should be replaced by "Laurales".

Figure 6: "Colored branch*e*s" in lower legend

Line 524: italic type for species name

Lines 529-563: The term "acetogenin" is not used consistently in biochemical literature and not exclusively for the acetogenins of Annonaceae. Originally, acetogenin means any metabolite formed via the acetate-polymalonate (polyketide) pathway, including fatty acids (which are obviously not confined to Annonaceae). It would thus be preferable to specify the unique Annonaceae acetogenins with a different/extended term. The term "tetrahydrofuran (THF) acetogenins" has been used before and would appear a suitable choice.

Line 586: italic type for "Arabidopsis" (2x)

Lines 627-660: Several switches between different types.

Line 637: See comment on lines 529-563 ("acetogenin") above.

Comments on the Quality of English Language

Some edits are required to clarify the text. See details in comments and suggestions for authors.

Reviewer 2 Report

Comments and Suggestions for Authors

This study is very important for elucidating the early stages of angiosperm evolution.

Author Response

Thanks for your approval of our paper.

Reviewer 3 Report

Comments and Suggestions for Authors

The manuscript presents the chromosome-level genome assembly of the Cherimoya cultivar 'Booth'. Using this genome sequence, the authors analyzed phylogenetic relationships, divergence time, and acetogenin biosynthetic genes, among others. This study contributes to the understanding of cherimoya genomics and its evolution, as well as the identification of gene clusters potentially responsible for acetogenin synthesis. Here are some comments and suggestions for improvement that should be addressed before considering publication of the manuscript:

All the results are obtained from DNA sequence analysis using bioinformatics methods. However, it is worth considering the use of RNA-seq and metabonomics to validate certain important genes and metabolites.

 It should provide a picture of the Cherimoya cultivar "Booth." Which tissues were used for DNA extraction and sequencing? What tissues do acetogenins mainly synthesize and distribute in?

 Since a comparative genome was analyzed between the two cherimoya genomes ("Fino de Jete" and “Booth”), what is the major difference in biological phonotype or economic traits between them?

 94 gene families have been significantly expanded in A. cherimola “Booth”, while 51 gene families significantly contracted (Figure 3A)”. With which species are these gene family expansions and contractions compared?

 Visualization of chromosomal differences: large difference in genome size between the two genomes, due to some major structural variations in Chr4 and Chr6.” “All chromosomes except for Chr4 are longer in “Fino de Jete” than in “Booth”.” In Fig. 4A, it is difficult to find the chromosome difference between the "Fino de Jete" and “Booth” genomes. Could you use chromosome alignment with gene synteny between the two genomes instead?

 The phylogenetic relationships among monocots, eudicots, and magnoliids is intriguing but somewhat confusing. It could be helpful to briefly summarize the current consensus (if one exists) before stating how the new genome might resolve outstanding conflicts.

Gb” and “gigabases” should be unified throughout the text.

Round 2

Reviewer 3 Report

Comments and Suggestions for Authors

All my questions and concerns have been addressed. I have no further questions.